# Phenolic Secondary Metabolites and Antiradical and Antibacterial Activities of Different Extracts of *Usnea barbata* (L.) Weber ex F.H.Wigg from Călimani Mountains, Romania

**DOI:** 10.3390/ph15070829

**Published:** 2022-07-04

**Authors:** Violeta Popovici, Laura Bucur, Cerasela Elena Gîrd, Antoanela Popescu, Elena Matei, Georgeta Camelia Cozaru, Verginica Schröder, Emma Adriana Ozon, Ancuța Cătălina Fița, Dumitru Lupuliasa, Mariana Aschie, Aureliana Caraiane, Mihaela Botnarciuc, Victoria Badea

**Affiliations:** 1Department of Microbiology and Immunology, Faculty of Dental Medicine, Ovidius University of Constanta, 7 Ilarie Voronca Street, 900684 Constanta, Romania; violeta.popovici@365.univ-ovidius.ro (V.P.); victoria.badea@365.univ-ovidius.ro (V.B.); 2Department of Pharmacognosy, Faculty of Pharmacy, Ovidius University of Constanta, 6 Capitan Al. Serbanescu Street, 900001 Constanta, Romania; antoanela.popescu@365.univ-ovidius.ro; 3Department of Pharmacognosy, Phytochemistry, and Phytotherapy, Faculty of Pharmacy, Carol Davila University of Medicine and Pharmacy, 6 Traian Vuia Street, 020956 Bucharest, Romania; 4Center for Research and Development of the Morphological and Genetic Studies of Malignant Pathology, Ovidius University of Constanta, CEDMOG, 145 Tomis Blvd., 900591 Constanta, Romania; sogorescuelena@gmail.com (E.M.); drcozaru@yahoo.com (G.C.C.); aschiemariana@yahoo.com (M.A.); 5Clinical Service of Pathology, Sf. Apostol Andrei Emergency County Hospital, 145 Tomis Blvd., 900591 Constanta, Romania; 6Department of Cellular and Molecular Biology, Faculty of Pharmacy, Ovidius University of Constanta, 6 Capitan Al. Serbanescu Street, 900001 Constanta, Romania; 7Department of Pharmaceutical Technology and Biopharmacy, Faculty of Pharmacy, Carol Davila University of Medicine and Pharmacy, 6 Traian Vuia Street, 020956 Bucharest, Romania; dumitru.lupuliasa@umfcd.ro; 8Department of Oral Rehabilitation, Faculty of Dental Medicine, Ovidius University of Constanta, 7 Ilarie Voronca Street, 900684 Constanta, Romania; aureliana.caraiane@365.univ-ovidius.ro; 9Department of Microbiology, Faculty of Medicine, Ovidius University of Constanta, 1 University Street, 900470 Constanta, Romania; mihaela.botnarciuc@365.univ-ovidius.ro

**Keywords:** *Usnea barbata* (L.) Weber ex F.H. Wigg extracts, phenolic secondary metabolites, usnic acid, polyphenols, DPPH free-radical scavenging activity, antibacterial activity

## Abstract

Phenolic compounds represent an essential bioactive metabolites group with numerous pharmaceutical applications. Our study aims to identify and quantify phenolic constituents of various liquid and dry extracts of *Usnea barbata* (L.) Weber ex F.H. Wigg (*U. barbata*) from Calimani Mountains, Romania, and investigate their bioactivities. The extracts in acetone, 96% ethanol, and water with the same dried lichen/solvent ratio (*w*/*v*) were obtained through two conventional techniques: maceration (*m*UBA, *m*UBE, and *m*UBW) and Soxhlet extraction (*d*UBA, *d*UBE, and *d*UBW). High-performance liquid chromatography with diode-array detection (HPLC-DAD) was performed for usnic acid (UA) and different polyphenols quantification. Then, the total phenolic content (TPC) and 2,2-diphenyl-1-picrylhydrazyl (DPPH) free-radical scavenging activity (AA) were determined through spectrophotometric methods. Using the disc diffusion method (DDM), the antibacterial activity was evaluated against Gram-positive and Gram-negative bacteria known for their pathogenicity: *Staphylococcus aureus* (ATCC 25923), *Streptococcus pneumoniae* (ATCC 49619), *Pseudomonas aeruginosa* (ATCC 27853), and *Klebsiella pneumoniae* (ATCC 13883). All extracts contain phenolic compounds expressed as TPC values. Five lichen extracts display various UA contents; this significant metabolite was not detected in *d*UBW. Six polyphenols from the standards mixture were quantified only in ethanol and water extracts; *m*UBE has all individual polyphenols, while *d*UBE shows only two. Three polyphenols were detected in *m*UBW, but none was found in *d*UBW. All *U. barbata* extracts had antiradical activity; however, only ethanol and acetone extracts proved inhibitory activity against *P. aeruginosa*, *S. pneumoniae*, and *S. aureus*. In contrast, *K. pneumoniae* was strongly resistant (IZD = 0). Data analysis evidenced a high positive correlation between the phenolic constituents and bioactivities of each *U. barbata* extract. Associating these extracts’ properties with both conventional techniques used for their preparation revealed the extraction conditions’ significant influence on lichen extracts metabolites profiling, with a powerful impact on their pharmacological potential.

## 1. Introduction

Phenolic compounds are essential plant secondary metabolites with numerous pharmaceutical applications [1]. As unique symbionts between fungi and algae, lichens are distinguished in the plants’ world by their specific secondary metabolites with phenolic structures (depsides, depsidones, dibenzofurans, anthraquinones, and xanthones) [2]. These constituents are deposited as crystals on fungal hyphae in the cortex or medulla; the different distribution in the thallus layers is correlated with their biological actions [3]. The lichen’s most significant pharmacological activities are antioxidant [4], antimicrobial [5], anticancer [6], photoprotective [7], and anti-inflammatory [8]. Therefore, they are considered important representatives with biopharmaceutical potential [9]. Due to remarkable antioxidant [10] and antibacterial [11] properties, lichens represent a promising source of protective [12,13,14] and antibiotic drugs [15,16,17].

With numerous pharmacological activities, the lichens of the genus *Usnea* (*Parmeliaceae*) are appreciated as powerful phytomedicines, used for therapeutical purposes for thousands of years [18]. The most known secondary metabolite in *Usnea* sp. is usnic acid—a phenolic compound with a dibenzofuran structure. As yellow crystals, it is found on cortex fungal hyphae, exhibiting a photoprotective action [19]. Usnic acid is found as a (+) enantiomer in *Usnea* lichens [20]. A valuable representative of this genus, known for its antioxidant [21], antibacterial [22], and photoprotective [7] effects, is *U. barbata*. Usnic acid is the main secondary metabolite responsible for its pharmacological potential [23]. The pharmaceutical applications of UA as an antibacterial agent are limited by its poor water solubility [24] and significant hepatotoxicity [25]. Therefore, the nanosystems with usnic acid must be able to increase its bio-disponibility, tolerance, and antibacterial effects [26]. Interesting nano-formulations were performed: liposomal UA-cyclodextrin inclusion complexes, which increase usnic acid solubility in water [27], glycosylated cationic liposomes, promoting usnic acid penetration in the bacterial biofilm matrix [28], and magnetic nanoparticles [29] with antimicrobial activity and antibiofilm activity against Gram-positive bacteria (*S. aureus* and *E. faecalis*) and Gram-negative ones (*P. aeruginosa*). Balaz et al. [30] recently proposed a bio-mechanochemical synthesis of silver nanoparticles using *U. antarctica* and other lichen species. Using AgNO_3_ (as a silver precursor) and lichens (as reduction agents), they performed techniques of mechanochemistry (ball milling) and obtained nanoparticles with an intense antibacterial effect against *S. aureus*. This described procedure overcomes the lichen secondary metabolites’ low solubility in water. Siddiqi et al. [31] demonstrated the antimicrobial properties of *U. longissima*-driven silver nanoparticles through the denaturation of ribosomes, leading to enzyme inactivation and protein denaturation, resulting in bacterial apoptosis.

*U. barbata* also contains bioactive polyphenols with pharmaceutical applications; different nanotechnologies were described to enhance their bioavailability and biocompatibility [32]. They can be used as nanoparticles to increase their antioxidant and antibacterial potential or other activities [33,34,35,36,37,38].

Numerous studies investigated the antibacterial effects of *Usnea* sp. Extracts—obtained through conventional and green extraction techniques—for pharmaceutical applications [39]. Thus, Tosun et al. [40] explored the antimycobacterial action of *U. barbata* fractions in petroleum ether, chloroform, methanol, and water. Bate et al. [41] studied the antibacterial activity of *U. articulata* and *U. florida* methanol macerates against MDR bacteria (*Staphylococcus* sp., *P aeruginosa*, *Salmonella* sp., and *E. coli*). Zizovic et al. [42] proved the strong antibacterial action of *U. barbata* supercritical fluid extracts (SFE). One year later, Ivanovic et al. [43] analyzed the influence of various extraction conditions (temperature, pressure) and pre-treatment methods on bactericidal effects against *S. aureus* strains. Basiouni et al. [44] evaluated the *U. barbata* sunflower oil extract inhibitory activity on bacterial strains isolates from poultry. In a previous study, Matvieva et al. [15] analyzed the antimicrobial properties of the ethanol, isopropanol, acetone, DMSO, and water extracts of *Usnea* sp against *S. aureus*, *B. subtilis*, and *E coli.*

We propose to investigate the antibacterial and antiradical properties of *U barbata* extracts in the same solvents, obtained by two low-cost and easy-to-use conventional techniques. Our study novelty consists of a comparative analysis of fluid and dry *U. barbata* extracts in ethanol, acetone, and water, obtained by maceration and Soxhlet extraction [34], determining their phenolic constituents and evaluating the free radical scavenging activity and antibacterial effects. Our results revealed that, despite the same ratio between the dried lichen and the solvent (*w*/*v*), all *U. barbata* extracts display significant differences in the phenolic metabolites’ diversity and amount due to extraction conditions, with a substantial impact on their bioactivities.

## 2. Results

### 2.1. Lichen Extracts

All data regarding the obtained *U. barbata* extracts are displayed in Table 1 and Appendix A.

Data from Table 1 show that the extraction temperature for liquid extracts was 20–22 °C, and their color varies from yellow (*m*UBA) to light brown (*m*UBE) and brown-reddish (*m*UBW).

At Soxhlet extraction, the temperature value increased from *d*UBA (55–60 °C) to *d*UBE (75–80 °C) and *d*UBW (95–100 °C). The highest yield (11.15%) was obtained for *d*UBE; its value decreased to 5.55% for *d*UBA and 1.76% for *d*UBW. Moreover, the dry extracts color changed from yellow-brown (*d*UBA) to light brown (*d*UBE) and dark brown-reddish (*d*UBW).

### 2.2. HPLC-DAD Determination of Usnic Acid Content

The usnic acid contents in all *U. barbata* extracts are displayed in Table 2.

All liquid extracts contain UA. Thus, *m*UBA had the highest UA content (211.9 mg/g extract equivalent to 21.19 mg/g dried lichen), following in decreasing order *m*UBE (0.257 mg/g, corresponding to 0.025 mg/g dried lichen) and *m*UBW (0.045 mg/g corresponding to 0.004 mg/g dried lichen). According to https://pubchem.ncbi.nlm.nih.gov/compound/Usnic-acid (accessed on 20 May 2022), usnic acid solubility significantly decreases in order: acetone > ethanol > water; these data can explain our results.

The chromatograms of usnic acid standard and *U. barbata* extracts in all three solvents are displayed in Figure 1.

Data from Table 2 show that only two dry extracts contain UA because in *d*UBW it was non-detected. Dry acetone extract contains UA of 241.773 mg/g, corresponding to 13.418 mg/g dried lichen. The usnic acid content in *d*UBE is 108.752 mg/g (12.125 mg/g dried lichen).

### 2.3. HPLC-DAD Determination of Polyphenols

The polyphenols contents are displayed in Table 3.

The chromatograms of *U. barbata* extracts are displayed in Figure 2, Figure 3, Figure 4 and Figure 5.

As can be seen, six polyphenols of the standard mixture were identified only in ethanol and water fluid extracts; their high solubility in polar solvents could justify their absence in acetone extracts (Table 3 and Figure 2 and Figure 3).

Of all six polyphenols identified in *m*UBE: caffeic acid (CA), *p*-coumaric acid (pCA), ellagic acid (EA), chlorogenic acid (ChA), gallic acid (GA), and cinnamic acid (CiA), only two (EA and GA) were found in *d*UBE, and three (pCA, ChA, and GA) in *m*UBW (Figure 2, Figure 3 and Figure 4). The common polyphenol for all three extracts is GA, with the highest content in *m*UBW (60.358 mg/g), followed by *m*UBE (27.487 mg/g) and *d*UBE (0.870 mg/g). Ellagic acid content is 230.819 mg/g in *m*UBE and 0.605 mg/g in *d*UBE (Table 3, Figure 4).

The common polyphenols for *m*UBW and *m*UBE were pCA and ChA; their amounts were higher in *m*UBW (0.749 and 0.627 mg/g) than *m*UBE (0.312 and 0.512 mg/g). The other two polyphenols—CA (0.414 mg/g) and CiA (17.948 mg/g)—were identified exclusively in *m*UBE (Table 3, Figure 2).

The *d*UBW chromatogram (Figure 5) shows three peaks at the following retention times (RT): 15.113 min, 15.642 min, and 16.091 min; these RT values differed from standard polyphenols’ ones. Their absence in *d*UBW could be due to their thermolability; the Soxhlet extraction involves prolonged heating for 8 h at 95–100 °C [45].

The polyphenols from the standard mixture were also non-detected in both *U. barbata* acetone extracts (Table 3) because their solubility is lower in this solvent than in ethanol or water.

### 2.4. Total Phenolic Content

It can be observed that the highest total phenolic content (TPC) values belong to dry *U. barbata* extracts (Table 4). The *d*UBA had the highest TPC (862.843 mg PyE/g); it is followed in decreasing order by *d*UBE (573.234 mg PyE/g) and *d*UBW (111.626 mg PyE/g). The TPC values in fluid extracts decreased in the following order: *m*UBE (276.603 mg PyE/mL), *m*UBA (220.597 mg PyE/mL), and *m*UBW (176.129 mg PyE/mL). TPC includes usnic acid, identified polyphenols, and unidentified phenolic constituents of each *U. barbata* extract.

### 2.5. Free-Radical Scavenging Activity Assay

The results are displayed in Table 4.

Data from Table 4 show that all *U. barbata* extracts have antiradical activity. This effect was higher for dry ethanol and acetone extracts (16.728% for *d*UBE, 15.471% for *d*UBA) than fluid ones (12.162% for *m*UBE, 11.146% for *m*UBA). Only for water extracts, the antiradical activity of *d*UBW (3.951%) is lower than the *m*UBW one (6.429%).

### 2.6. Antibacterial Activity

The obtained results proved that the negative control (DMSO 0.1%) has no inhibitory effect on the bacteria tested (IZD = 0 mm). Only *U. barbata* extracts in acetone and ethanol inhibited bacterial strains’ growth. (Appendix A). Neither UBWs have any inhibitory effect on the tested bacteria (IZD = 0 mm).

Given that usnic acid is the major secondary metabolite of the genus *Usnea*, we considered this phenolic compound as a positive control. For the optimal interpretation of the obtained IZD values, we used two bactericidal antibiotics with different mechanisms of action and breakpoints: ofloxacin (OFL) and ceftriaxone (CTR).

The data displayed in Table 5 show the IZD values (mm) for all *U. barbata* extracts, UA, and standard antibiotics drugs (OFL and CTR).

Therefore, comparing the IZD values of the *U. barbata* extracts to those of both standard antibiotics on *S. aureus*, none had antibacterial action (IZD = 11.00–13.66 mm). Only usnic acid has an IZD (16.33 mm) in the “I” range of ofloxacin (17–15 mm); this means that antibacterial activity on *S. aureus* is dose dependent. Compared to ceftriaxone, the IZD value for UA belongs to the resistance range (<20 mm).

*S. pneumoniae* is sensitive to all *U. barbata* extracts as well as to usnic acid (IZD = 17.33–18.67 mm) when IZD values are compared to ofloxacin (S ≥ 16 mm *). However, it could be considered resistant when IZD values were compared to CTR (S ≥ 26 mm *).

Among Gram-negative bacteria, *P. aeruginosa* proves the highest sensitivity; all lichen extracts showed antibacterial action on *P. aeruginosa* (IZD = 16.77–20.33 mm), compared to ofloxacin (*S* ≥ 16 mm *). Only ethanol extracts (IZD = 20.00–20.33 mm) had an antibacterial effect related to ceftriaxone (S ≥ 18 mm *); the others are active in a dose-dependent manner (I = 17–15 mm). Contrariwise, no *U. barbata* extract inhibited the growth of *K. pneumoniae* colonies (IZD = 0 mm).

Considering the data registered in Table 5, we calculated the antibacterial activity index (AI), reporting the IZD values (mm) of lichen extracts to the ones of the standard antibiotic drugs [47]. It can be noted that dry and fluid *U. barbata* acetone and ethanol extracts had similar inhibitory effects (Table 6).

The presence of similar bioactive secondary metabolites, the fluid extracts used after solvent evaporation, and the additional presence of the polyphenols known for their strong antibacterial action could explain the results registered in Table 5 and Table 6. Thus, UA had the highest inhibitory activity on *S. aureus*, showing a dose-dependent antibacterial effect and the highest AI values; the following are the extracts with a high usnic acid content, respectively UBA. *U. barbata* ethanol extracts show the lowest inhibitory effect because usnic acid is known for its highest inhibition levels on *S. aureus*; both UBEs have lower UAC values than the corresponding UBAs ones (Table 5).

On *S. pneumoniae* and *P. aeruginosa*, the lichen extracts in ethanol indicated the most significant inhibitory levels. Antibacterial activities of individual polyphenols could justify these results. They showed an antibacterial action against *S. pneumoniae* similar to ofloxacin. On *S. pneumoniae*, the AI values compared to OFL are statistically different from those linked to CTR (Table 6). In this case, for all *U. barbata* extracts, AI ≥ 0.912, proving that their antibacterial activity is similar to OFL. Against *P. aeruginosa*, *m*UBA and *d*UBA reported AI values higher than OFL (AI > 1) and similar to CTR (AI ≥ 0.952) (Table 6).

### 2.7. Data Analysis

We obtained *U. barbata* extracts performing two easy-to-use and low-cost conventional techniques mentioned in Romanian Pharmacopoeia X [48]: maceration for fluid extracts and Soxhlet extraction for dry ones. They have been one of the most used extraction procedures for herbal bioactive compounds [49]. According to the green chemistry concept, the solvents used for lichen extraction are “preferable,” having low toxicity and significant safety [50]. Our entire study’s data were synthesized in Table 7.

From the beginning, the same ratio—1:10 (*w*/*v*) between dried lichen and solvent—was maintained for all extracts. The fluid extracts were obtained at room temperature (20–22 °C). The Soxhlet extraction was performed by prolonged heating, and the requested temperature values registered in Table 7 were maintained for 8 h.

The phenolic metabolites contents were strongly influenced by extraction conditions, as shown in Table 7. Usnic acid content and TPC significantly increase in acetone and ethanol dry extracts than in fluid ones; UBAs have higher UAC and TPC than UBEs. The *m*UBW had the lowest TPC and UAC. However, after 8 h of Soxhlet extraction at 100 °C, *d*UBW shows diminished TPC values and no UAC.

The individual polyphenols were quantified only in ethanol and water *U. barbata* extracts. The *m*UBE contains all six polyphenols (CA, CiA, pCA, EA, GA, and ChA) and *m*UBW—only three (pCA, GA, and ChA). Regarding the corresponding dry extracts, in *d*UBE only two polyphenols (EA and GA) were found in lower content than *m*UBE; *d*UBW has no polyphenols.

These detailed aspects could be explained in the first step by the solubility differences of phenolic compounds in each extraction solvent. Polyphenols are soluble in polar solvents (ethanol, water); however, they are affected by prolonged heating [45]; thus, it can justify their decreasing or absence in the dry extracts after Soxhlet extraction for 8 h at 75–80 °C (*d*UBE) and 95–100 °C (*d*UBA). The lowest solubility of usnic acid in water underlies the minimal UAC value in *m*UBW. The high temperature of extraction (100 °C for 8 h) affects usnic acid stability; thus, the absence of UA in *d*UBW could be justified. According to https://www.biocrick.com/Usnic-acid-BCN4306.html (accessed on 2 May 2022), usnic acid storage requests desiccation and freezing (−20 °C); this information supports our results.

On the other hand, it can be seen that the dry extracts are obtained with a considerably low yield. When all UAC values are reported to the dried lichen amount used for each extract preparation, 2.119% corresponds to *m*UBA and only 1.341% for *d*UBA.

#### Principal Component Analysis

Principal component analysis (PCA) was performed for all *U. barbata* liquid and dry extracts and variable parameters—according to the correlation matrix from Appendix A—and illustrated in Figure 6.

The PCA-Correlation circle from Figure 6a explains 84.40% of the data variances [51] and correlates the lichen extracts metabolites with their bioactivities. It can be observed that the horizontal axis (PC1) is linked to pCA, GA, and ChA, usnic acid content, TPC, AA, and antibacterial activities. PC2 is associated with CA, EA, and CiA. Figure 6a shows that UA moderately correlates with the lichen extracts bioactivities: AA (*r* = 0.626, *p* > 0.05), S.a. (*r* = 0.728, *p* > 0.05), S.p. (*r* = 0.625, *p* > 0.05), and P.a. (*r* = 0.545, *p* > 0.05). TPC displays a good positive correlation with AA (*r* = 0.822, *p* < 0.05) and the moderate ones with antibacterial activities—*r* values decrease from 0.693 (S.a.) to 0.603 (S.p.) and 0.563 (P.a.), *p* > 0.05. We can also observe that AA is highly correlated with antibacterial activities—*r* values are 0.923 (S.a.), 0.900 (S.p.), and 0.897 (P.a.), *p* < 0.05—because in both effects involve the phenolic metabolites, with their phenolic -OH groups (Figure 6a). The individual polyphenols are insignificantly (positively or negatively) correlated with both bioactivities for all lichen extracts because these compounds were quantified only in three *U. barbata* extracts (Figure 6a).

The PCA-Correlation circle from Figure 6b explains 79.38% of the data variances and correlates the lichen extracts metabolites with extraction temperature. All parameters (except TPC, *r* = 0.209) are negatively correlated with the temperature (*p* > 0.05). The temperature values moderately correlate with pCA (*r* = −0.587), ChA (*r* = 0.652) and GA (*r* = 0.594). Other variable parameters reported a low negative correlation with extraction temperature (detailed data in Appendix A). Usnic acid with temperature registered the lowest negative correlation (*r* = −0.042).

The lichen extracts’ phytoconstituents significantly influence their pharmacological potential. Hence, we explored the metabolites content to explain the differences in the obtained results regarding antiradical and antibacterial effects. Then, we determined the correlations between these bioactivities and phenolic metabolites quantified in each lichen extract. All data are displayed in Figure 7, Figure 8, Figure 9 and Figure 10 and detailed in Appendix A.

In *m*UBE, all quantified phenolic secondary metabolites significantly correlate with DPPH free radical scavenging ability (AA, *r* ≥ 0.930) and antibacterial activities (Figure 7).

As expected, Figure 7 shows a high correlation (*r* = 0.999, *p* < 0.05) between pCA and TPC and AA and S.a. Ellagic acid remarkably correlates with AA (*r* = 0.930, *p* > 0.05) and all antibacterial effects—*r* value decreases from 0.996 (P.a.) to 0.989 (S.p.) and 0.930 (S.a.), *p* > 0.05. The phenolic compounds correlate with the inhibitory effect against *S. aureus* registering the highest correlation index values (*r* ≥ 0.930, *p* > 0.05), followed by the one against *P. aeruginosa* (*r* = 0.817–0.996, *p* > 0.05) and *S. pneumoniae* (in the most cases, a moderate correlation, *r* = 0.655–0.867, *p* > 0.05). UA shows the highest correlation with S.a. (*r* = 0.945, *p* > 0.05), followed by P.a. (*r* = 0.817, *p* > 0.05) and S.p. (*r* = 0.655, *p* > 0.05). Moreover, AA is considerably correlated with all antibacterial activities, S.a. (*r* = 0.999, *p* < 0.05), P.a. (*r* = 0.961, *p* > 0.05) and S.p. (*r* = 0.866, *p* > 0.05).

In *d*UBE, we identified two polyphenols (gallic acid and ellagic acid) and UA. The phenolic metabolites remarkably correlate with both bioactivities (*r* ≥ 0.848, *p* </> 0.05, Figure 8).

Data illustrated in Figure 8 highlight the strongest correlation (*r* = 0.999, *p* < 0.05) between phenolic compounds (EA, GA, and TPC) and AA and P.a. On *P. aeruginosa*, the powerful action of ellagic acid and gallic acid is due to phenolic compound general mechanisms and biofilm inhibition [52]. The same correlation (*r* = 0.999, *p* < 0.05) can be noticed between UA and S.a.; UA is a valuable antibacterial compound against *S. aureus* and, as a positive control, had a dose-dependent antibacterial effect. Both activities—AA and P.a.—are also highly correlated (*r* = 0.999, *p* < 0.05).

TPC of *m*UBA and *d*UBA are positively correlated with antibacterial effects (Figure 9). In *m*UBA, TPC correlates with S.a. (*r* = 0.999, *p* < 0.05); it also corellates with S.p. and P.a. in *d*UBA. UA moderately corellates with S.p. (*r* = 0.515, *p* < 0.05) in *m*UBA and S.a. in *d*UBA (*r* = 0.723, *p* > 0.05). In both UBAs, UA (*r* = 0.827 and 0.884, *p* > 0.05) and TPC (*r* = 0.996 and 0.978, *p* > 0.05) display a high correlation with AA. These correlations are evidenced in Figure 9. Furthermore, in both UBAs, DPPH free-radical scavenging activity and antibacterial effects are strongly correlated (*r* = 0.906, 0.962 and 0.970, *p* > 0.05, Figure 9).

These correlations associated with the bio-activities of all quantified metabolites could explain the similar inhibitory activity on bacterial strains growing of both *U barbata* extracts in ethanol and acetone. Moreover, in these extracts, all phenolic metabolites could synergistically act.

The PCA-correlation circle for UBWs is displayed in Figure 10.

Data from Figure 10 show that usnic acid (*r* = 0.910, *p* > 0.05) and individual polyphenols—pCA (*r* = 0.951, *p* > 0.05), GA and ChA (*r* = 0.999, *p* < 0.05) highly correlate with AA in liquid water extract. Furthermore, in both UBWs, TPC show a powerful correlation with AA (*r* = 0.995, and 0.961, *p* < 0.05). However, because the phenolic compounds with known antibacterial action were extracted in water in minimal quantities, both UBWs did not exhibit any inhibitory effect on bacteria tested (IZD = 0).

Our study deeply analyzed six *U. barbata* extracts in three solvents, from the description of extraction conditions to phenolic constituents’ determination and the evaluation of their biological activities. A detailed data analysis was performed on the correlations between phenolic metabolites and biological activities for each *U. barbata* extract, aiming to explain the obtained results. We correlated phenolic metabolites with antiradical and antibacterial activities and with extraction temperature for all six *U. barbata* extracts. The extraction temperature’s significant role was highlighted by comparing the liquid and dry extracts in the same solvent. Thus, we evidenced the strong influence of the extraction temperature on phenolic metabolites diversity and content and, consequently, the strong impact on antiradical and antibacterial activities.

Correlating and interpreting all data, we made each lichen extract characterization, highlighting the similar and different properties compared to the others (Figure 11).

Figure 11a shows that the fluid UBE (obtained at room temperature) contains UA in a low content and all six polyphenols in an appreciable amount. It can be noticed that CA, EA, and CiA are associated exclusively with *m*UBE; moreover, it shares ChA, GA, and pCA with *m*UBW. Individual polyphenols contribute considerably to the *m*UBE’s TPC value (Figure 11a). These constituents could synergistically act, leading to their significant antiradical and antibacterial potential (Figure 11b). The Soxhlet extraction at 75–80 °C significantly diminished the polyphenols content; thus, *d*UBE reported low concentrations of only two polyphenols (EA and GA, Figure 11a). Moreover, UA and other phenolic secondary metabolites were resistant to prolonged heating and detected in dry acetone extract (Figure 11a). Therefore, *d*UBE shows a higher AA than *m*UBE and similar antibacterial effects. The fluid water extract (*m*UBW) shows the lowest content of phenolic metabolites compared to other macerates. It contains three individual polyphenols (pCA, GA, ChA) and usnic acid (Figure 11a). Despite the antibacterial properties of all phenolic constituents, their content is too low, and *m*UBW does not inhibit bacterial strains’ growth; it has only moderate antiradical activity (Figure 11b). The prolonged heating at 100 °C during Soxhlet extraction diminished phenolics content; UA and individual polyphenols from *m*UBW were not detected in *d*UBW (Figure 11a), and AA decreased.

Both acetone extracts (*m*UBA and *d*UBA) have the same metabolites (UA and TPC) and bioactivities (Figure 11b); the temperature and yield have a quantitative influence, increasing UAC and TPC in *d*UBA. Therefore, AA augments and antibacterial properties are similar. In Figure 11a,b, both UBAs and *d*UBE are positioned at low distances; both UBWs are located in the same quarter of the PCA–biplot, thus evidencing their similar properties.

## 3. Discussion

The low yields associated with diminished UAC in dried lichen can also be observed in other studies on *U. barbata* extracts obtained in various conditions [42,43,53,54]. The most relevant data are displayed in Table 8.

The data from Table 8 indicate that the UAC (%) in dried lichen generally decreases directly proportional to the extraction yield when the same solvent is used.

The usnic acid chemical structure strongly relates to *U. barbata* antiradical and antibacterial activities [22]. Due to protonophore and uncoupling action, all three phenolic OH groups of UA are essential [55], leading to bacterial membrane potential dissipation, associated with bacterial colonies growing inhibition. Maciag-Dorszynska et al. [56] proved that usnic acid produces a rapid and strong inhibition of nucleic acids synthesis in Gram-positive bacteria (*S. aureus* and *B. subtilis*). It could also inhibit Group A Streptococcus (*Streptococcus pyogenes*) biofilm formation [57], reducing biofilm biomass and depleting the biofilm-forming cells’ proteins and fatty acids. Sinha et al. [58] proved that UA could act synergistically with norfloxacin and modify *S. aureus* methicillin-resistant (MRSA) drug resistance. This effect involves efflux pump inhibition, oxidative stress induction, and down-regulation of peptidoglycans and fatty acids biosynthesis. These mechanisms alter membrane potential and perturb cell respiration and metabolic activity.

The polyphenols could synergistically act with usnic acid and other secondary metabolites in *U. barbata* extracts’ antiradical and antibacterial activities. The antibacterial effects of polyphenols implicate various mechanisms. Thus, Lou et al. [59] proved that the *p*-coumaric acid bactericidal effect against *S. aureus* and *S. pneumoniae* involves irreversible permeability changes in bacterial cell walls and binding to bacterial genomic DNA; as a result, it occurs cell function inhibition followed by bacteria cell death. Caffeic acid (CA) acts as an antibacterial drug through various mechanisms; it produces cell membrane depolarization and disruption, reduces the respiratory activity of bacteria, decreases efflux activity, affects intracellular redox processes, donates protons, and increases intracellular acidity [34]. Moreover, CA proved to have an appreciable inhibitory effect against *S. aureus* (IZD = 12 mm) [34]. Cinnamic acid (CiA) preferentially acts against Gram-negative bacteria (*P. aeruginosa*), determining cell membrane damage, affecting its lipidic profile, and leading to protein loss and denaturation [60]. Chlorogenic acid (ChA) antibacterial mechanisms involve outer cell membrane bounding and disrupting, intracellular potential exhausting, and loss of cytoplasm macromolecules, leading to cell death [61]. On *S. pneumoniae*, ChA inhibits a key virulence factor (neuraminidase) [62]. Gallic acid (GA) has a significant antibacterial effect against Gram-positive bacteria (*S. aureus*, *Streptococcus* sp.), increasing their ability to accept electrons. On Gram-negative bacteria, this property could decrease, indicating that GA is an electrophilic compound interacting with bacterial surface components [63,64,65]. Ellagic acid (EA) acts on *S. aureus* damaging the bacteria cell membrane, leading to significant leakage of proteins and nucleic acids. Its antibacterial activity could inhibit protein synthesis, inducing great morphological changes in bacterial cell structure [66]. Both phenolic acids (GA and EA) also proved bactericidal effects against *P. aeruginosa* [52]. In encapsulated form, their antibacterial potential could increase [38].

Numerous researchers analyzed the antibacterial activity of *U. barbata* and *Usnea* sp.; generally, their results were similar to those obtained in our study [39]. The sensibility of Gram-positive bacteria to usnic acid and various *Usnea* sp. extracts is most known. Idamokoro et al. [67] analyzed the effect of *U. barbata* extracts in methanol and ethyl-acetate against 13 isolated *Staphylococcus* sp. involved in cow mastitis. They evidenced ethyl-acetate extract’s lower inhibitory activity than methanol ones. On *S. aureus*, they reported an IZD value = 14 mm for methanol extract, similar to our *d*UBA (IZD = 13.66 mm). Mesta et al. [68] indicated the IZD values of 12 mm—for *U. ghatensis* ethanol extract 15 mg/mL against *S. aureus*—and 18 mm—for *U. undullata* ethanol extract 15 mg/mL on *S. pneumoniae*; both values are similar to those for *m/d*UBE obtained in the present study. In a previous study [69], we evaluated the antibacterial activity of *U. barbata* liquid extracts against two other *Streptococcus* sp. (*S. oralis* and *S. intermedius*) isolated from the oral cavity. Those obtained IZD values proved that *m*UBE had a stronger action for both *Streptococcus* sp. than *m*UBA; *m*UBW did not show any inhibitory effect. No inhibitory effects (IZD = 0) displayed the extracts of *U. pectinata*, *U. coraline*, and *U. baileyi* in methanol and dichloromethane against *K. pneumoniae* [5]. The methanol extracts of *U. articulata* (IZD = 28 mm) and *U. florida* (IZD = 18 mm) highlighted a remarkable antibacterial action against *P. aeruginosa* [41]. *U. florida* extract in methanol also proved significant activity on *S. aureus* (IZD = 30). Boisova et al. [70] optimized the conditions of UA SFE extraction from *U. subfloridana* (for 80 min, at a temperature of 85 °C and pressure of 150 atm). Their obtained extract proved an intense antibacterial activity against *S. aureus*.

## 4. Materials and Methods

### 4.1. Materials

Our study’s chemicals, reagents, and standards were of analytical grade. Usnic acid standard 98.1% purity, phenolic standards (Z-resveratrol, caffeic acid, E-resveratrol, chlorogenic acid, ferulic acid, gallic acid, ellagic acid, p-coumaric acid, vanillin, 3-methyl gallic acid, cinnamic acid) were purchased from Sigma (Sigma-Aldrich Chemie GmbH., Taufkirchen, Germany). Folin–Ciocâlteu reagent, Pyrogallol, DPPH, acetone, and ethanol were supplied by Merck (Merck KGaA, Darmstadt, Germany).

The bacterial lines were obtained from Microbiology Department, S.C. Synevo Romania SRL, Constanta Laboratory, in partnership agreement No 1060/25.01.2018 with the Faculty of Pharmacy, Ovidius University of Constanta. Culture media Mueller–Hinton agar simple and one with 5% defibrinated sheep blood were supplied by Thermo Fisher Scientific, GmbH, Dreieich, Germany.

### 4.2. Lichen Extracts

*U. barbata* was harvested from Călimani Mountains, Romania (47°28′ N, 25°10′ E, 900 m altitude) in March 2021. The lichen was dried at a constant temperature below 25 °C in an airy room, protected from the sunlight. After drying, the obtained herbal product was preserved for a long time in the same conditions for use in subsequent studies. The lichen was identified using standard methods by the Department of Pharmaceutical Botany of the Faculty of Pharmacy, Ovidius University of Constanta. A voucher specimen (*Popovici 3/2021 Ph/UOC*) [71] can be found at the Department of Pharmacognosy, Faculty of Pharmacy, Ovidius University of Constanta.

The dried lichen was ground in an LM 120 laboratory mill (PerkinElmer, Waltham, MA, USA) and passed through the no. 5 sieve [19]. The obtained moderately fine lichen powder (particle size ≤ 315 μm) was subjected to extraction in acetone, 96% ethanol, and water (dried lichen: solvent ratio (*w*/*v*) = 1:10) using two conventional techniques.

The first procedure was maceration—three samples of 10 g ground dried lichen were extracted with 100 mL solvent (water, acetone, and 96% ethanol) in a dark place at room temperature (20–22 °C) for 10 days, with manual shaking 3–4 times/day. The resulting extractive solutions were filtered and made up of a 100 mL volumetric flask with each solvent. These fluid extracts (*m*UBA, *m*UBE, and *m*UBW) were preserved in dark-glass recipients with sealed plugs in the same conditions until processing.

The second one was Soxhlet extraction for 8 h, with the temperature values around each solvent’s boiling point. Thus, three samples of 20 g ground dried lichen were refluxed at Soxhlet for eight hours with 200 mL of each solvent. Acetone and 96% ethanol were evaporated at the rotary evaporator TURBOVAP 500 (Caliper Life Sciences Inc, Hopkinton, MA, USA). Then, these extracts were kept for 16 h in a chemical exhaust hood for optimal solvent evaporation. After filtration with filter paper, UBW was concentrated on a Rotavapor R-215 with a vacuum controller V-850 (BÜCHI Labortechnik AG, Flawil, Switzerland), and lyophilized with a freeze-dryer Christ Alpha 1-2L (Martin Christ Gefriertrocknungsanlagen GmbH, Osterode am Harz, Germany) connected to a vacuum pump RZ 2.5 (VACUUBRAND GmbH, Wertheim, Germany) [72]. All these dry extracts (*d*UBA, *d*UBE, *d*UBW) were transferred in sealed-glass containers and preserved in freezer (Sirge^®^ Elettrodomestici—S.A.C. Rappresentanze, Torino, Avigliana, Italy) at −18 °C [73] until processing.

### 4.3. HPLC-DAD Determination of Usnic Acid Content

A previously validated HPLC-DAD method was adapted for quantifying usnic acid [53].

#### 4.3.1. Equipment and Chromatographic Conditions

This analytic method used an Agilent 1200 HPLC (Agilent Technologies, Santa Clara, CA, USA) with a G1311 quaternary pump, Agilent 1200 G1315B diode array detector (DAD), G1316 thermostatted column compartment, G1322 vacuum degassing system, G1329 autosampler.

The system has a Zorbax C18 analytical column 150 mm/4.6 mm; 5 µm (Agilent Technologies, Santa Clara, CA, USA). As a mobile phase, isocratic methanol: water: acetic acid = 80:15:5 was selected for 6 min per run, at an injection volume of 20 µL at a flow rate = 1.5 mL/min. The oven temperature was established at 25 °C, and the detection was performed at 282 nm.

#### 4.3.2. Sample, Blank, Standard Solutions

All requested solutions were prepared using acetone as a solvent. The standard was usnic acid dissolved in acetone at concentrations of 2.5, 5, 10, 20, 50 µg/mL, with which the calibration curve (Appendix A) was drawn (y = 39.672x − 3.8228; *R*^2^ = 0.999). Each dilution was injected 6 times (20 µL) in the chromatographic system, and the obtained retention time value was 4.463 ± 0.008 min.

#### 4.3.3. Data Processing

Data processing was achieved using the Waters Empower 2 chromatography data software with ICS 1.05 (Waters Corporation, Milford, MA, USA).

### 4.4. HPLC-DAD Determination of Polyphenols

The polyphenols quantification was achieved using a standardized HPLC method. It was described by the USP 30-NF25 monograph and previously validated [74].

#### 4.4.1. Equipment and Chromatographic Conditions

The Agilent HPLC-DAD system was the analytical platform, with the same Zorbax C18 column, 150 mm 4.6 mm; 5 µm. As a mobile phase, two solutions were used: solution A: 0.1% phosphoric acid and solution B: acetonitrile, with gradient elution, at 22 min per run, with the same injection volume and flow The temperature was set at 35 °C and the detection was performed at UV 310 nm.

#### 4.4.2. Sample, Blank, Standard Solutions

The standard solutions were 70% methanol solutions with various concentrations of: Z-resveratrol (0.22 mg/mL), caffeic acid (0.36 mg/mL), E-resveratrol (0.37 mg/mL), chlorogenic acid (0.37 mg/mL), ferulic acid (0.38 mg/mL), gallic acid (0.39 mg/mL), ellagic acid (0.40 mg/mL), p-coumaric acid (0.41 mg/mL), vanillin (0.42 mg/mL), 3-methyl gallic acid (0.51 mg/mL), cinnamic acid (0.58 mg/mL). The retention time values (minutes), established after 6 injections with each standard were displayed in Appendix A; all phenolic standards have *R*^2^ values > 0.99, as admissibility condition. The samples were the *U. barbata* extracts in different solvents (their preparation was mentioned in the Section 4.2).

### 4.5. Total Phenolic Content

The total phenolic content was determined using Folin–Ciocâlteu reagent through a spectrophotometric method detailed in a previous study [53]. Pyrogallol was selected as the standard, the TPC values being calculated as mg of pyrogallol equivalents (PyE) per gram extract.

### 4.6. DPPH Free-Radical Scavenging Activity Assay

The *U. barbata* extracts free radical scavenging activity (AA) was determined spectrophotometrically through the DPPH free-radical scavenging assay previously described [19].

### 4.7. Antibacterial Activity

The antibacterial effects were evaluated by an adapted disc diffusion method (DDM) from the Clinical and Laboratory Standard Institute (CLSI) [75], previously described [76].

#### 4.7.1. Microorganisms and Media

We obtained all bacteria strains from the American Type Culture Collection (ATCC). Their identification was performed at the Department of Microbiology and Immunology, Faculty of Dental Medicine, Ovidius University of Constanta. The Gram-positive bacteria were *S. aureus* (ATCC 25923) and *S. pneumoniae* (ATCC 49619); the Gram-negative ones were *Pseudomonas aeruginosa* (ATCC 27853) and *K. pneumoniae* (ATCC 13883). As a culture medium for all bacterial strains, Mueller–Hinton agar was used.

#### 4.7.2. Inoculum Preparation

We prepared the bacterial inoculum using the direct colony suspension method (CLSI). Thus, we obtained a 0.9% saline suspension of bacterial colonies selected from a 24 h agar plate, according to the 0.5 McFarland standard, with around 10^8^ CFU/mL (CFU—colony-forming unit).

#### 4.7.3. Lichen Samples Preparations

The fluid extracts were subjected to solvent evaporation in the rotary evaporator TURBOVAP 500. These concentrated extracts were kept for 2 h in a chemical exhaust hood for each optimal solvent evaporation. Then, all *U. barbata* extracts were redissolved in 0.1% DMSO [77], obtaining a final solution of 15 mg/mL concentration.

The dry lichen extracts were dissolved in 0.1% DMSO, resulting in 15 mg/mL concentration solutions.

#### 4.7.4. Disc Diffusion Method

The 15 mg/mL lichen extracts in 0.1% DMSO were applied on Whatman^®^ filter paper discs (6 mm, Merk KGaA, Darmstadt, Germany). The negative control was the solvent (0.1% DMSO); UA of 15 mg/mL in 0.1% DMSO was the positive control for all extracts. We impregnated each filter paper disc with 10 µL control and sample solutions. The standard antibiotic discs (6 mm) with ofloxacin 5 µg and ceftriaxone 30 µg (Oxoid, Thermo Fisher Scientific GmbH, Dreieich, Germany) were selected for antimicrobial activity evaluation. These blank discs were stored in a freezer at −14 °C and incubated for 2 h before analysis at room temperature.

Each inoculum was applied over the entire surface of the plate with the suitable culture media using a sterile cotton swab. After 15 min of drying, the filter paper discs were applied to the inoculated plates; they were incubated at 37 °C for 24 h.

#### 4.7.5. Reading Plates

Circular zones of a microorganism growing inhibition around several discs could be observed, examining the plates after 24 h incubation. The results of the disc diffusion assay are expressed in the inhibition zone diameter (IZD) measured in mm. These IZD values quantify bacterial strains’ susceptibility levels after 24 h incubation [78].

#### 4.7.6. Interpretation of Disc Diffusion Method results

Usnic acid and *U. barbata* extracts’ IZD were compared to the IZD values of the positive controls represented by the blank antibiotic discs, ofloxacin 5 ug and ceftriaxone 30 ug [78]. In DDM, IZD values inversely correlate with minimum inhibitory concentrations (MIC) from standard dilution tests. According to CLSI [78], the interpretive categories are as follows: susceptible (“S”), intermediate—dose-dependent susceptibility (“I”), and resistant (“R”) [46].

#### 4.7.7. Activity Index

The activity index (AI) [47] is calculated using the following formula:(1)$$\mathrm{AI}=\frac{\mathrm{IZD}\;\mathrm{sample}}{\mathrm{IZD}\;\mathrm{standard}}\;$$
where IZD sample—inhibition zone diameter for each *U. barbata* extract, and IZD standard—inhibition zone diameter for each antibacterial drug, used as standard.

### 4.8. Data Analysis, Software

All analyses were accomplished in triplicate, and the results are expressed as the mean (*n* = 3) ± SD, calculated by Microsoft 365 Office Excel (Redmond, Washington, DC, USA). The *p*-values were calculated with the one-way ANOVA test; when the *p*-value was <0.05, the differences between the obtained mean values were considered significant. The principal component analysis (PCA) [51] was performed using XLSTAT 2022.2.1. by Addinsoft (New York, NY, USA) [79].

## 5. Conclusions

Our study analyzed the phenolic constituents and bioactivities of six *U. barbata* lichen extracts obtained through two low-cost conventional techniques widely used in pharmaceutical laboratories. Despite the same ratio between the dried lichen and the solvent (*w*/*v*), all lichen extracts displayed significant differences regarding the phenolic metabolites’ diversity and amount due to extraction conditions, with a substantial impact on their bioactivities. All *U. barbata* extracts show antiradical activity; the antibacterial study proves that the *U. barbata* extracts in acetone and ethanol obtained through both methods considerably inhibit bacterial colony growth. Both Gram-positive bacteria and *P. aeruginosa* of Gram-negative ones reveal the highest sensibility.

Our results suggest that further research could extend the antibacterial studies, exploring their effects on other bacteria species. Future studies could optimize both extraction processes to obtain *U. barbata* extracts with valuable bioactivities for potential pharmaceutical applications.

## Figures and Tables

**Figure 1 pharmaceuticals-15-00829-f001:**
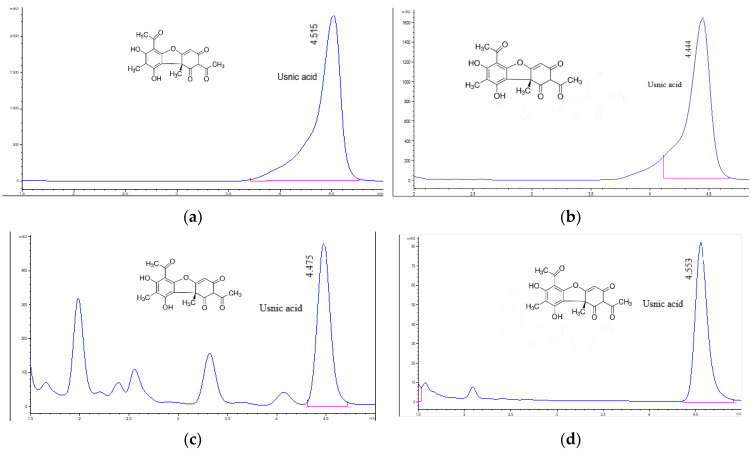
Chromatograms of usnic acid standard (**a**), *m*UBA (**b**), *m*UBE (**c**), *m*UBW (**d**). The red lines mark the significant peak areas.

**Figure 2 pharmaceuticals-15-00829-f002:**
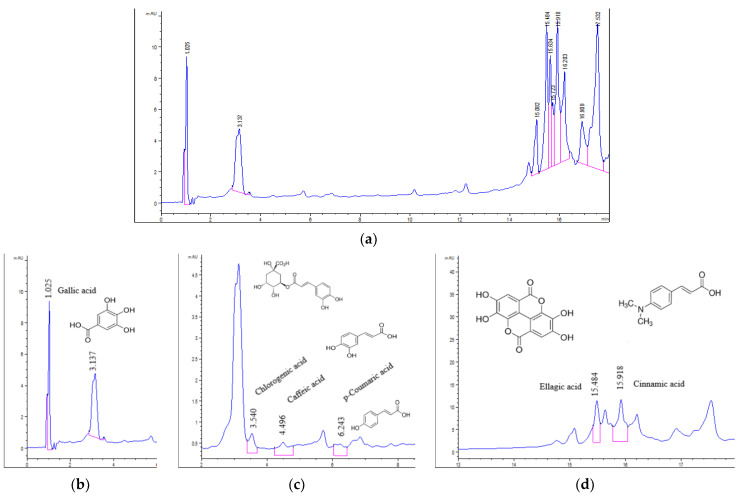
Chromatograms of *m*UBE (**a**); polyphenols in *m*UBE: gallic acid (**b**); chlorogenic, caffeic, and *p*-coumaric acids (**c**); ellagic and cinnamic acids (**d**). The red lines mark the significant peak areas, UBE—*U. barbata* ethanol extract, *m*—macerate.

**Figure 3 pharmaceuticals-15-00829-f003:**
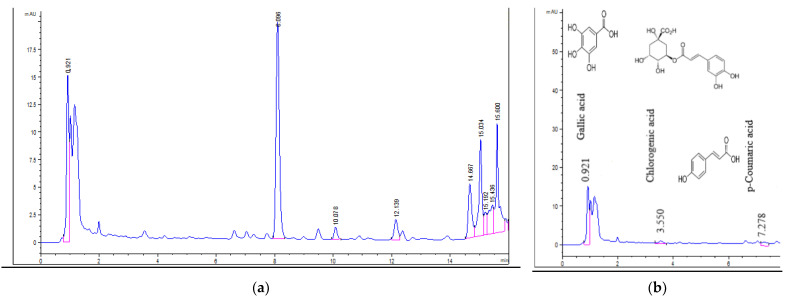
Chromatograms of *m*UBW (**a**); polyphenols in *m*UBW (**b**). The red lines mark the significant peak areas; UBW—*U. barbata* water extract, *m*—macerate.

**Figure 4 pharmaceuticals-15-00829-f004:**
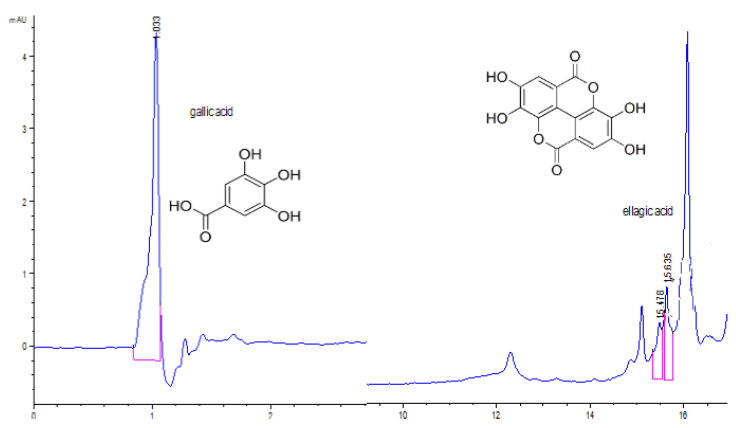
Gallic acid and ellagic acid in *d*UBE. The red lines mark the significant peak areas.

**Figure 5 pharmaceuticals-15-00829-f005:**
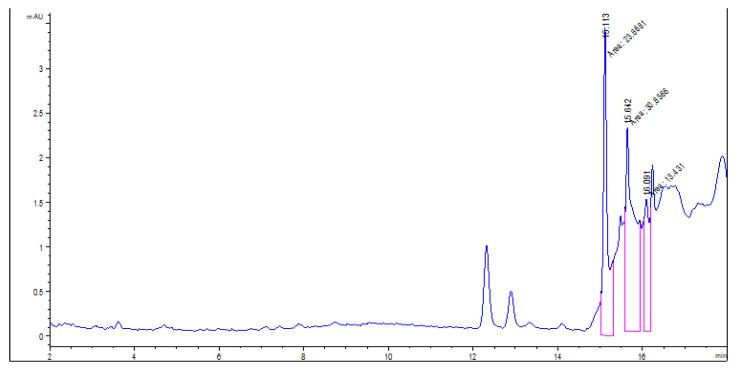
Chromatogram of *d*UBW. The red lines mark the significant peak areas.

**Figure 6 pharmaceuticals-15-00829-f006:**
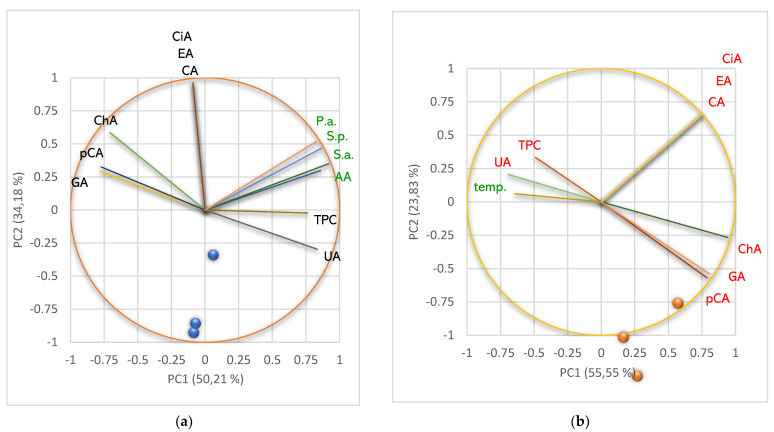
Principal component analysis (PCA): PCA-Correlation circle between phenolic metabolites and bioactivities of *U. barbata* extracts (**a**); PCA-Correlation circle between phenolic metabolites and extraction temperature (**b**). pCA—*p*-coumaric acid, ChA—chlorogenic acid, CA—caffeic acid, CiA—cinnamic acid, EA—ellagic acid, GA—gallic acid, UA—usnic acid, TPC—total phenolic content, AA—antiradical activity, P.a.—inhibitory activity against *P. aeruginosa*, S.a.—inhibitory activity against *S. aureus*, S.p.—inhibitory activity against *S. pneumoniae*, temp—extraction temperature.

**Figure 7 pharmaceuticals-15-00829-f007:**
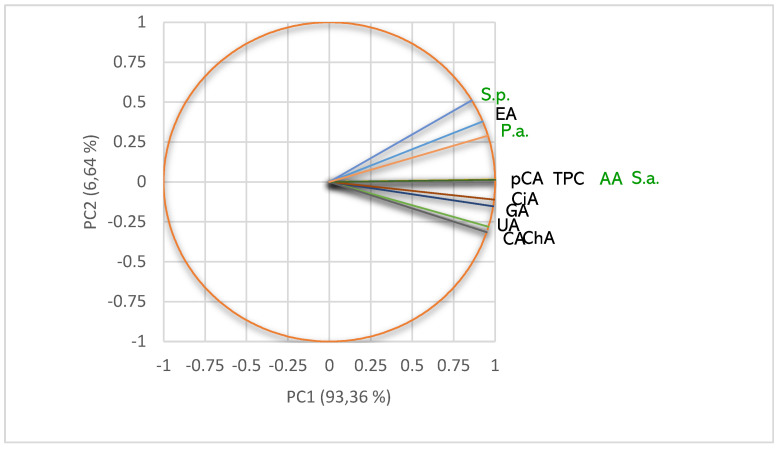
PCA-Correlation circle between TPC, UA, and individual polyphenols in *m*UBE and antibacterial and antiradical activities. *m*UBE—*U. barbata* liquid ethanol extract, pCA—*p*-coumaric acid, ChA—chlorogenic acid, CA—caffeic acid, CiA—cinnamic acid, EA—ellagic acid, GA—gallic acid, UA—usnic acid, TPC—total phenolic content, AA—antiradical activity, P.a.—inhibitory activity against *P. aeruginosa*, S.a.—inhibitory activity against *S. aureus*, S.p.—inhibitory activity against *S. pneumoniae*.

**Figure 8 pharmaceuticals-15-00829-f008:**
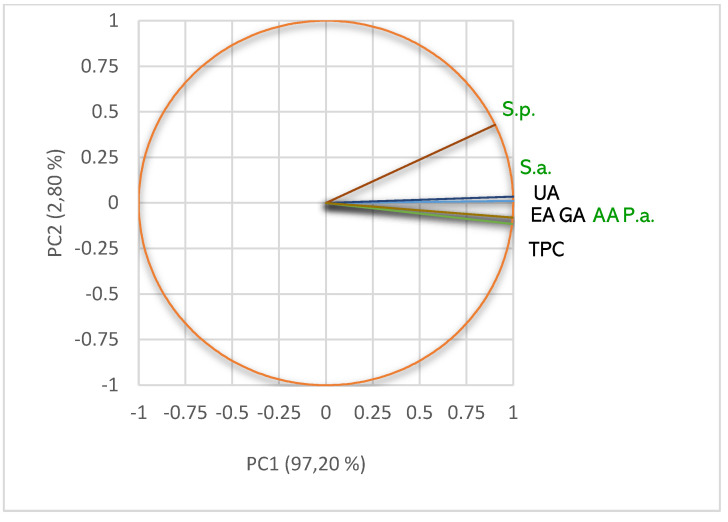
PCA-Correlation circle between TPC, UA, and individual polyphenols in *d*UBE and antibacterial and antiradical activities. *d*UBE—*U. barbata* dry ethanol extract, EA—ellagic acid, GA—gallic acid, UA—usnic acid, TPC—total phenolic content, AA—antiradical activity, P.a.—inhibitory activity against *P. aeruginosa*, S.a.—inhibitory activity against *S. aureus*, S.p.—inhibitory activity against *S. pneumoniae*.

**Figure 9 pharmaceuticals-15-00829-f009:**
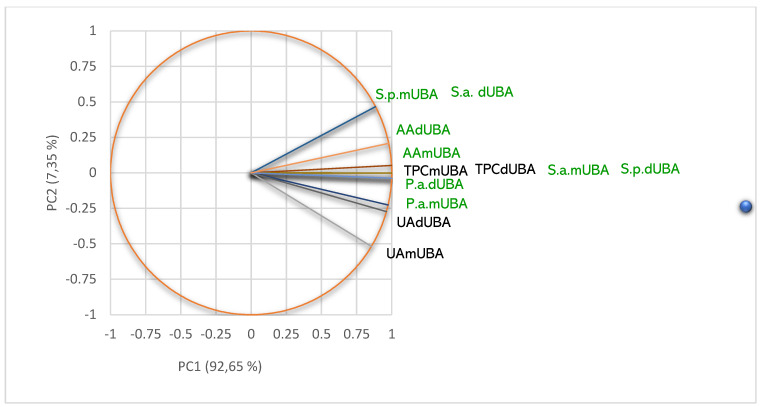
PCA-Correlation circle between TPC, UA, and individual polyphenols in *m*UBA and *d*UBA and antibacterial and antiradical activities; UBA—*U. barbata* acetone extract, *m*—macerate, *d*—dry; UA—usnic acid, TPC—total phenolic content, AA—antiradical activity, P.a.—inhibitory activity against *P. aeruginosa*, S.a.—inhibitory activity against *S. aureus*, S.p.—inhibitory activity against *S. pneumoniae*.

**Figure 10 pharmaceuticals-15-00829-f010:**
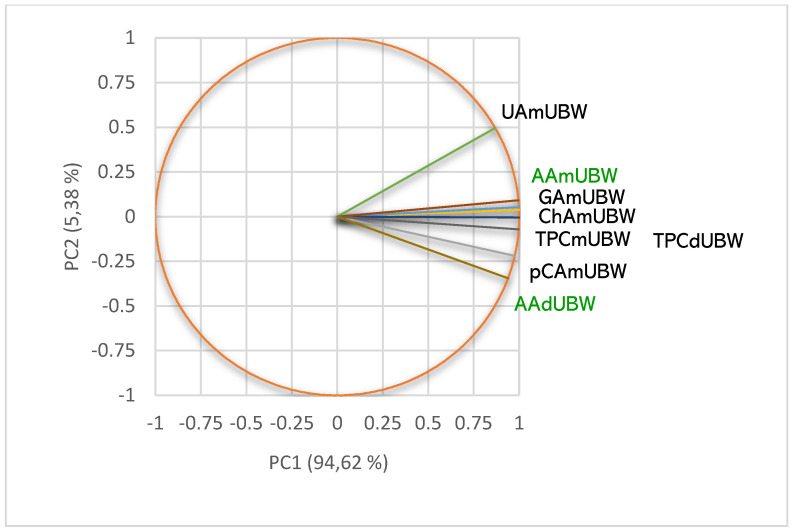
PCA-Correlation circle between TPC, UA, and individual polyphenols in *m*UBW and only between TPC in *d*UBW and antiradical activities; UBW—*U. barbata* water extract, *m*—macerate, *d*—dry; pCA—*p*-coumaric acid, ChA—chlorogenic acid, GA—gallic acid, UA—usnic acid, TPC—total phenolic content, AA—antiradical activity.

**Figure 11 pharmaceuticals-15-00829-f011:**
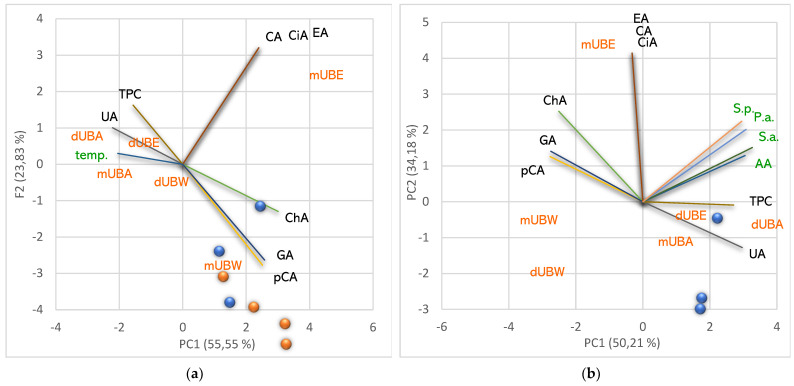
Characterization of *U. barbata* extracts by positioning each lichen extract according to its phenolic metabolites correlated with temperature (**a**) and bioactivities (**b**). pCA—*p*-coumaric acid, ChA—chlorogenic acid, CA—caffeic acid, CiA—cinnamic acid, EA—ellagic acid, GA—gallic acid, UA—usnic acid, TPC—total phenolic content, AA—antiradical activity; P.a.—inhibitory activity against *P. aeruginosa*, S.a.—inhibitory activity against *S. aureus*, S.p.—inhibitory activity against *S. pneumoniae;* temp—extraction temperature. UBA—*U. barbata* acetone extract, UBE—*U. barbata* ethanol extract, UBW—*U. barbata* water extract; *m*—macerate, *d*—dry extract.

**Table 1 pharmaceuticals-15-00829-t001:** Extraction conditions and *U. barbata* extracts color.

Extraction Solvent	*U. barbata* Extract	Temperature of Extraction (°C)	Yield (%)	*U. barbata* Extract’s Color
Acetone	*d*UBA	55–60	5.55 ^b^	Yellow-brown
	*m*UBA	20–22	n/a	Yellow
Ethanol	*d*UBE	75–80	11.15 ^a^	Light brown
	*m*UBE	20–22	n/a	Light brown
Water	*d*UBW	95–100	1.76 ^c^	Dark brown-reddish
	*m*UBW	20–22	n/a	Brown reddish

UBA—*U. barbata* acetone extract, UBE—*U. barbata* ethanol extract, UBW—*U. barbata* water extract; *m*—macerate, *d*—dry extract, n/a—not applicable. The yield values followed by superscript letters are statistically significant (*p* < 0.05).

**Table 2 pharmaceuticals-15-00829-t002:** Usnic acid content in fluid and dry *U. barbata* extracts.

*U. barbata* Extract		UAC	
		mg/g Lichen Extract	mg/g Dried Lichen
Acetone	*m*UBA	211.900 ± 0.002 ^b^	21.190 ^f^
	*d*UBA	241.830 ± 0.172 ^a^	13.418 ^g^
Ethanol	*m*UBE	0.257 ± 0.002 ^d^	0.025 ^i^
	*d*UBE	108.742 ± 0.703 ^c^	12.125 ^h^
Water	*m*UBW	0.045 ± 0.002 ^e^	0.004 ^j^
	*d*UBW	ND	n/a

UAC—usnic acid content, UBA—*U. barbata* acetone extract, UBE—*U. barbata* ethanol extract, UBW—*U. barbata* water extract; *m*—macerate, *d*—dry extract, ND—non-detected, n/a—not applicable; the mean values followed by superscript letters are statistically significant (*p* < 0.05).

**Table 3 pharmaceuticals-15-00829-t003:** Polyphenols contents in *U. barbata* fluid and dry extracts in ethanol and water.

*U. barbata* Extracts	*m*UBE	*m*UBW	*d*UBE	*d*UBW
Polyphenols	Polyphenols Content mg/g Lichen Extract			
Caffeic acid (CA)	0.414 ± 0.005	ND	ND	ND
p-coumaric acid (pCA)	0.312 ± 0.001 ^b^	0.749 ± 0.049 ^a^	ND	ND
Ellagic acid (EA)	230.819 ± 0.264 ^c^	ND	0.605 ± 0.007 ^d^	ND
Chlorogenic acid (ChA)	0.512 ± 0.006 ^f^	0.627 ± 0.006 ^e^	ND	ND
Gallic acid (GA)	27.487 ± 0.459 ^h^	60.358 ± 0.363 ^g^	0.870 ± 0.008 ^k^	ND
Cinnamic acid (CiA)	17.948 ± 0.114	ND	ND	ND

UBA—*U. barbata* acetone extract, UBE—*U. barbata* ethanol extract, UBW—*U. barbata* water extract; *m*—macerate, d—dry; pCA—*p*-coumaric acid, ChA—chlorogenic acid, CA—caffeic acid, CiA—cinnamic acid, EA—ellagic acid, GA—gallic acid, ND—non-detected; the mean values followed by superscript letters are statistically significant (*p* < 0.05).

**Table 4 pharmaceuticals-15-00829-t004:** Total phenolic content (TPC) and free-radical scavenging activity of *U. barbata* extracts.

*U. barbata* Extract		TPC (mg PyE/g Extract)	DPPH-Free Radical Scavenging%
Acetone	*m*UBA	220.597 ± 24.527 ^d^	11.146 ± 0.577 ^k^
	*d*UBA	862.843 ± 33.727 ^a^	15.471 ± 0.629 ^h^
Ethanol	*m*UBE	276.603 ± 15.025 ^c^	12.162 ± 0.396 ^j^
	*d*UBE	573.234 ± 42.308 ^b^	16.728 ± 0.284 ^g^
Water	*m*UBW	176.129 ± 24.169 ^e^	6.429 ± 0.286 ^l^
	*d*UBW	111.626 ± 11.132 ^f^	3.951 ± 0.297 ^m^

TPC—total phenolic content, UBA—*U. barbata* acetone extract, UBE—*U. barbata* ethanol extract, UBW—*U. barbata* water extract; *m*—macerate, *d*—dry extract, mg PyE—mg equivalents pyrogallol. The mean values followed by superscript letters are statistically significant (*p* < 0.05).

**Table 5 pharmaceuticals-15-00829-t005:** Antibacterial activity of *U. barbata* extracts.

Bacteria		*S. aureus*	*S. pneumoniae*	*P. aeruginosa*	*K. pneumoniae*
Inhibition Zone Diameter—IZD (mm)					
UA		16.33 ± 0.82	17.33 ± 0.47	16.67 ± 0.47	0
*Liquid extracts*					
*m*UBA		12.00 ± 0.82 ^b^	17.67 ± 0.47	17.33 ± 1.25	0
*m*UBE		11.00 ± 0.82 ^d^	18.67 ± 0.47	20.33 ± 1.70	0
*m*UBW		0	0	0	0
*Dry extracts*					
*d*UBA		13.66 ± 0.47 ^a^	18.00 ± 1.63	17.00 ± 1.63	0
*d*UBE		12.33 ± 1.25 ^c^	18.33 ± 0.47	20.00 ± 1.63	0
*d*UBW		0	0	0	0
*Standard antibacterial drugs inhibitory activity*					
OFL 5		26.33 ± 1.70	19.00 ± 1.63	19.33 ± 1.70	30.00 ± 0.82
CTR 30		25.00 ± 2.45	32.33 ± 2.05	21.00 ± 2.16	32.33 ± 2.49
*Standard antibacterial drugs breakpoints **					
*Ofloxacin*					
OFL 5	S *	≥18 *	≥16 *	≥16 *	≥16 *
	I *	17–15 *	15–13 *	15–13 *	15–13 *
	R *	≤14 *	≤12 *	≤12 *	≤12 *
*Ceftriaxone*					
CTR 30	S *	≥21 *	≥26 *	≥18 *	≥23 *
	I *			17–15 *	22–20 *
	R *	≤20 *	≤25 *	≤14 *	≤19 *

UBA—*U. barbata* acetone extract, UBE—*U. barbata* ethanol extract, UBW—*U. barbata* water extract; *m*—macerate, *d*—dry extract, UA—usnic acid (positive control), * Data adapted from CLSI breakpoints analyzed by Humphries et al. [46]; OFL—ofloxacin, CTR—ceftriaxone; 5, 30 µg—the antibiotic amount from the standard antibiotic disc; S—sensitivity, I—intermediate (dose-dependent action), R—resistance. The superscripts letters noted the statistically significant IZD mean values (*p* < 0.05).

**Table 6 pharmaceuticals-15-00829-t006:** Antibacterial activity index of *U. barbata* extracts and UA compared to both standard antibiotic drugs.

Bacteria	AI Values (Adim)					AB
	*m*UBA	*d*UBA	*m*UBE	*d*UBE	UA	
*S. aureus*	0.455	0.519	0.417	0.468	0.620	OFL5
	0.480	0.546	0.440	0.490	0.693	CTR30
*S. pneumoniae*	0.930 ^a^	0.947 ^a^	0.982 ^a^	0.964 ^a^	0.912 ^a^	OFL5
	0.546 ^b^	0.556 ^b^	0.577 ^b^	0.566 ^b^	0.536 ^b^	CTR30
*P. aeruginosa*	0.896	0.879	1.051	1.034	0.862	OFL5
	0.825	0.809	0.968	0.952	0.793	CTR30

AI—antibacterial activity index, adim—without measure unit, UBA—*U. barbata* acetone extract, UBE—*U. barbata* ethanol extract, *m*—macerate, *d*—dry extract, UA—usnic acid, AB—standard antibiotic drug, OFL—ofloxacin, CTR—ceftriaxone. 5, 30 µg—the antibiotic amount from the standard antibiotic disc. The AI values noted with superscripts letters are statistically significant (*p* < 0.05).

**Table 7 pharmaceuticals-15-00829-t007:** Characteristics of *U. barbata* extracts in ethanol, acetone, and water obtained by two different conventional techniques, regarding the extraction conditions, phenolic metabolites, and bioactivities.

*U. barbata* Extract		*m*UBE	*d*UBE	*m*UBA	*d*UBA	*m*UBW	*d*UBW
*Extraction conditions*							
Solvent		96% ethanol		Acetone		Water	
Ratio (w/v)		1:10					
Temperature (°C)		20–22	75–80	20–22	55–60	20–22	95–100
Yield (%)			11.150		5.550		1.760
*Phenolic metabolites (mg/g extract*)							
TPC		276.603	573.234	220.597	862.843	176.129	111.626
UA	mg/g extract	0.257	108.74	211.190	241.830	0.045	
	% in dried lichen	0.002	1.212	2.119	1.341	0.0004	
CA		0.414					
pCA		0.312				0.749	
EA		230.820	0.605				
GA		27.487	0.870			60.358	
CiA		17.948					
ChA		0.513				0.627	
*Antibacterial activity—IZD (mm*)							
S.a.		11.000	12.330	12.000	13.670		
S.p.		18.670	18.330	17.670	18.000		
P.a.		20.330	20.000	17.330	17.000		
*DPPH free radical scavenging activity (%*)							
AA		12.162	16.728	11.146	15.471	6.429	3.951

pCA—*p*-coumaric acid, ChA—chlorogenic acid, CA—caffeic acid, CiA—cinnamic acid, EA—ellagic acid, GA—gallic acid, UA—usnic acid, TPC—total phenolic content, AA—antiradical activity, P.a.—inhibitory activity against *P. aeruginosa*, S.a.—inhibitory activity against *S. aureus*, S.p.—inhibitory activity against *S. pneumoniae;* UBA—*U. barbata* acetone extract, UBE—*U. barbata* ethanol extract, UBW—*U. barbata* water extract; *mUBE*, *mUBA*, *mUBW*—obtained by maceration; *d*UBE, *d*UBA, *d*UBW—obtained by Soxhlet extraction.

**Table 8 pharmaceuticals-15-00829-t008:** Various *U. barbata* extracts with different extraction conditions correlated with the yield and usnic acid content expressed as mg/g extract, and % UA in dried lichen.

*U. barbata* Extract	Extraction Solvent	Conditions of Extraction				Yield %	UAC (mg/g in Extract)	% UA in Dried Lichen *
		Pressure (Mpa)	Temperature (°C)	CO_2_ Pressure (m^3^/kg)	Pretreat- ment			
UBDEA ^a^	Ethyl acetate		75–80			6.27	376.73	2.362
UB-SFE ^b^	99% CO_2_	30	60			0.38	594.80	2.226
UB-SFE ^b^	99% CO_2_	30	40			0.60	364.90	2.190
UBO ^c^	Canola oil		22				0.915	2.162
UBDA ^a^	Acetone		55–60			6.36	282.78	1.798
UBDE ^a^	96% ethanol		75–80			12.52	127.21	1.592
UBDM ^a^	Methanol		65			11.29	137.60	1.553
UB-SFE ^d^	99% CO_2_	50	40	992	CM	2.28	545	1.243
					RM	1.67	585	0.977
					UM + RGD	1.50	645	0.968
		30	40	911	UM	1.27	617	0.806
					UM + RGD	1.46	423	0.618
					UM	0.85	648	0.551
					RM	0.78	634.5	0.481
					CM	0.86	558.1	0.479

UB SFE—*U. barbata* extract obtained by supercritical fluid extraction with CO_2_, UBDEA—*U. barbata* dry extract in ethyl acetate, UBDA—*U. barbata* dry extract in acetone, UBDE—*U. barbata* dry extract in ethanol, UBDM—*U. barbata* dry extract in methanol, UBO—*U. barbata* extract in canola oil, RM—roller mill; UM—ultra-centrifugal mill; CM—cutting mill; RGD—rapid gas decompression. * Data registered in decreasing order; superscript letters evidenced the data adapted from: ^a^ [53], ^b^ [42], ^c^ [54], ^d^ [43].

## Data Availability

Data are contained within the article.

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
