# Peer review of "Phenolic Secondary Metabolites and Antiradical and Antibacterial Activities of Different Extracts of Usnea barbata (L.) Weber ex F.H.Wigg from Călimani Mountains, Romania"

_pharmaceuticals, 2022, doi:10.3390/ph15070829_

Round 1

Reviewer 1 Report

Interesting manuscript, authors should rethink the introduction part, much information can be moved to the discussion section. Did the authors consider cytotoxicity analysis of the extracts?

Author Response

Please, see the attachment!

Reviewer 2 Report

The work of Popovici and coworkers evaluates the properties of acetone, ethanol and aqueous extracts from Usnea barbata obtained by two different extraction methods. Antioxidant and antimicrobial properties were determined.

However, the main and most significant element in the paper that qualifies for improvement is the introduction and discussion.

The discussion and the introduction need to be significantly rewritten - the introduction is too long, some elements could be moved to the discussion. The discussion itself contains unnecessary descriptions of results, which should be placed in the results chapter. Thus all the graphs and figures should be moved to the chapter above. The discussion would need to be re-written to allow comparison of results from the selected topic and their evaluation.

In subsection 2.5 there is an error in the numbering of the table

Line 182, 183 - dots

Author Response

Please, see the attachment!

Reviewer 3 Report

In the manuscript entitled “Phenolic Secondary Metabolites and Antiradical and Antibacterial Activities of Different Extracts of Usnea barbata (L.) Weber ex F.H.Wigg from Călimani Mountains, Romania” the authors investigated phenolic constituents and bioactivities of Usnea barbata extracts. They demonstrated the impact of extraction conditions on the bioactivity of the extracts. The manuscript is well written and is easy to follow.

There are some minor issues that should be fixed:

Throughout the text change “bacteria inoculum” to “bacterial inoculum”.

Lines 48-49: Rewrite the sentence.

Line 121: Authors should add following citations:

-       Molecules 202025, 2947.

-        Plants, 2020, 9(12), 1680. https://doi.org/10.3390/plants9121680

Lines 146-156: Shorten this paragraph as it contains the full description of the article including the results and repeats the abstract.

Line 256: Change to “Table 4”, instead of “Table 2”.

Line 273: Change the table so that the values are all on the same line, as it is difficult to keep track of where something belongs.

Line 371: Change from “1;10” to “1:10”.

Line 380: Change U. barbata to italic.

Line 380: “In”, instead of “un”.

Author Response

Please, see the attachment!

Round 2

Reviewer 2 Report

Manuscript has been revised